

# What's for dinner this time?: DNA authentication of "wild mushrooms" in food products sold in the USA

W. Dalley Cutler II*, Alexander J. Bradshaw* and Bryn T.M. Dentinger

Natural History Museum of Utah & School of Biological Sciences, University of Utah, Salt Lake City, UT, United States

* These authors contributed equally to this work.

Corresponding author
Bryn T.M. Dentinger,
bryn.dentinger@gmail.com

## ABSTRACT

Mushrooms have been consumed by humans for thousands of years, and while some have gastronomic and nutritional value, it has long been recognized that only select species of mushrooms are suitable for consumption. Adverse health effects of consuming poisonous mushrooms range from mild illness to death. Many valuable edible mushrooms are either impractical or unable to be grown commercially, requiring them to be harvested from the wild. In the U.S., products containing these wild-collected mushrooms are often sold with the nonspecific and undefined label "wild mushrooms," although in some cases particular species are listed in the ingredients. However, the ambiguity of the definition of "wild mushrooms" in foods makes it impossible to know which species are involved or whether they are truly wild-collected or cultivated varieties. As a consequence, any individual adverse reactions to consuming the mushrooms in these products cannot be traced to the source due to the minimal regulations around the harvest and sale of wild mushrooms. For this study, we set out to shed light on what species of fungi are being sold as "wild mushrooms" using DNA metabarcoding to identify fungal contents of various food products acquired from locally sourced grocers and a large online retail site. Twenty-eight species of mushroom were identified across 16 food products, ranging from commonly cultivated species to wild species not represented in global DNA databases. Our results demonstrate that "wild mushroom" ingredients often consist entirely or in part of cultivated species such as the ubiquitous white and brown "button" mushrooms and portabella (*Agaricus bisporus*), oyster (*Pleurotus* spp.) and shiitake (*Lentinula edodes*). In other cases truly wild mushrooms were detected but they were not always consistent with the species on the label. More alarmingly, a few products with large distribution potential contained species whose edibility is at best dubious, and at worst potentially toxic.

## INTRODUCTION

Food authentication is an important service necessary to ensure the health and safety of consumers. Methods to authenticate foods have benefited from advances in molecular identification techniques, which have shown on multiple occasions that food producers are

not always honest about the contents of their products. For example, genomic and proteomic data have been used to show the presence of horse and pork in foods sold as beef or game meat (*Brooks et al., 2017*), and revealed widespread fraud in sushi products (*Lowenstein, Amato & Kolokotronis, 2009*) and other seafoods (*e.g.*, (*Ho et al., 2020*)). Contamination and substitution has also been found to be common in the herbal product market (*Newmaster et al., 2013*; *Speranskaya et al., 2018*). Yet, despite an increasing interest in food authentication, foods labeled as containing "wild mushrooms" have received relatively little attention.

Wild edible mushrooms are harvested worldwide for consumption and trade in local communities and, increasingly, at a global scale through international markets (*Boa, 2004*; *Dentinger & Suz, 2014*; *Li et al., 2021*). Many of these species cannot be grown commercially and thus must be collected from the wild. Although wild edible mushrooms, such as porcini (*Boletus edulis* and allies), are not considered to be poisonous, rare allergic responses following ingestion have been documented (*e.g.*, *Fischer et al., 2017*) and unintentional adulteration with exogenous chemicals can occur through environmental contamination or during processing. In fact, in 2010, nicotine levels in mushrooms exported from China were found to exceed the maximum residue levels set by the European Food Safety Authority (*Cavalieri, Bolzoni & Bandini, 2010*). Even without exogenous contamination, many species of mushroom have never been systematically examined for their edibility, especially those that originate from regions where fungi are severely under-documented (*Li et al., 2021*). Instead, the edibility status of wild mushrooms relies on traditional knowledge that may not translate effectively in the global market. For example, some mushrooms require special preparation techniques to remove toxins prior to consumption (*Niksic, Klaus & Argyropoulos, 2016*), but these specialized preparation techniques are not typically found on the labels. Moreover, collection of edible fungi from the wild requires in-depth knowledge of which fungi are edible and how to distinguish them from ones that are not, knowledge that is not trivial to gain. Incorrect identification followed by consumption can be inconsequential; however, some fungi are notoriously poisonous and many others are not tolerated well by most people, especially when eaten raw. Consuming inedible species of mushroom can lead to mild symptoms such as mild discomfort and illness, or far more seriously, death. Contributions from inexperienced wild mushroom collectors increases the probability for misidentified, and potentially hazardous, mushrooms to end up in the food supply chain. Moreover, knowledge of the ranges, population sizes, genetic diversity and heterogeneity, ecological requirements, and the impact of harvesting on their growth and survival are lacking for all wild mushrooms, making it impossible to properly regulate their harvest for sustainability and conservation. Therefore, insufficient knowledge of the taxonomic composition, geographic origins, processing techniques, and conservation status renders consumption of foods containing wild-collected mushrooms potentially a risky business, for both human health and the environment.

Policies that regulate the collection of wild mushrooms for personal consumption and commercial trade vary widely. In some parts of Europe, a license to collect edible wild mushrooms is required (*De Roman, 2010*). In addition, local experts may be available in an

official capacity to assist inexperienced collectors (*Von Hagen et al., 1998*). Still, guidelines and legislation on commercialization of wild collected mushrooms varies widely amongst EU countries (*Peintner et al., 2013*). At the level of the EU, legislation on food safety is enforced through the European Food Safety Authority. In the U.S., commercial sale of wild mushrooms is regulated by state governments. Of the 49 states that responded to a National Survey of State Regulation of Wild Mushroom Foraging for Retail Sale, only 31 had regulations in place (*Nair, 2016*). Of these 31 states, the majority (94%) did not require notification of health authorities prior to selling wild mushrooms. In most of the states that do regulate wild mushroom food products, it is required that certification or licensing be obtained to ensure vendors are qualified to correctly distinguish edible from nonedible species. For example, to sell wild mushrooms to food establishments in Minnesota, you must be a Certified Wild Mushroom Harvester, which requires documentation of completing a mushroom identification course at an accredited college, university or mycological society (https://www.mda.state.mn.us/food-feed/certified-wild-mushroom-harvester). These regulations are in place to protect consumers from poisoning as a result of misidentification. Vendors that obtain proper licensing or certifications are able to charge a premium due to the increased labor and logistical difficulties associated with the regulations. These regulatory costs coupled with the inherent difficulty of finding and collecting wild mushrooms at a commercial scale, can drive up the prices for wild-collected mushrooms and create a market for less expensive sources that can be sourced globally and distributed widely and efficiently through online retailers. To our knowledge, no regulations pertaining to the online sale of wild edible mushrooms exist in the U.S.

Cultivated species of mushroom can be used in place of wild species at a fraction of the cost, but not without losing the higher value consumers place on truly wild mushrooms for their variety of flavors and textures, and their relative rarity and limited seasonal availability. Because the term "wild mushroom" is not regulated in the U.S., its application may be inconsistent across the food industry. Unscrupulous (or uneducated) marketers can even use the label when only cultivated mushrooms are involved, which is a common problem (personal observation). But the extent to which this fraud is being committed, or the relative proportion of cultivated and truly wild mushrooms in these products, has rarely been examined (*Jensen-Vargas & Marizzi, 2018*; *Raja et al., 2017*; *Loyd et al., 2018*). In addition, online markets are much less regulated and are designed to sell products to consumers worldwide, which can have far-reaching consequences. For example, a recent investigation by the *Wall Street Journal* into the large online retailer Amazon revealed thousands of products for sale that were unsafe, banned or mislabeled (*Berzon, Shifflett & Scheck, 2019*). For wild-collected products such as foods and medicines, online retailers that facilitate international sourcing and distribution also create the possibility for companies to over-exploit the potentially delicate ecologies of wild fungi through overharvesting or unorthodox harvesting methods that can result in habitat degradation (*Norvell, 1995*; *IUCN, 2020*). Not only do most edible fungi lack formal protection, there are no international regulations in place to ensure that consumers receive wild mushrooms from qualified collectors. In fact, the lack of regulation of foods labeled as

containing wild mushrooms creates an opportunity for any type of fungus to make it into food products without any oversight. Although this currently represents a relatively small proportion of commercially available food products, non-animal derived sources of protein like those provided by edible fungi are of increasing interest, and the potential health consequences of misidentifications or fraudulent practices are very real. The combined effect of poor scientific documentation and lack of regulation can even yield surprising discoveries: one recent study identified three new species of porcini mushrooms from a packet of dried porcini purchased at a grocer in London, U.K. (*Dentinger & Suz, 2014*).

In an effort to shed light on what types of fungi are being sold as "wild mushrooms", we set out to identify what species are present in commonly available wild mushroom products. Using food products obtained from grocery stores in the Salt Lake City metropolitan area and a large, U.S.-based online retailer, we applied an amplicon metagenomic approach to identify mushroom species and compared them with what is advertized on the product label.

## METHODS

### Sampling

Sixteen different food products labeled as containing wild mushrooms, including dried mushrooms, powdered mushrooms, soups, pasta sauces, and flavor enhancers, were purchased from local supermarkets in Salt Lake City, Utah, USA, as well as from a large, U.S.-based online retailer (Fig. 1, Table 1). Specimen vouchers have been deposited in the fungarium at the Natural History Museum of Utah, which is part of the Garrett Herbarium (UT). To assess reproducibility of the data, we made replicates of the samples in two ways: (1) duplicate amplifications from the same DNA extract (indicated with "a" and "b" in the sample IDs) and (2) separate DNA extractions from the same sample (indicated by "1" and "2" in the sample IDs). We also processed three mushroom fragments from one packet independently to confirm identifications from the metabarcoding analysis with Sanger sequencing.

### DNA extraction, PCR & sequencing

Samples were processed differently depending on their type. For samples composed of pieces of dried mushrooms, as many morphologically distinct pieces as could be identified visually were sampled to maximize the potential diversity sampled from the packet. These pieces were then combined and flash frozen in liquid nitrogen before grinding them into a fine powder with a mortar and pestle. Wet pasta sauce samples were processed by separating the liquids and solids with a mesh strainer and collecting the large fragments that remained. These fragments were homogenized by placing them in 2.0 mL screw cap tubes containing a single 3.0 mm and five to ten 1.5 mm stainless steel beads, and shaking them in a BeadBug microtube homogenizer (#Z763713, Sigma, St. Louis, MO, USA) for 120 seconds at speed setting 400. Samples that were already powdered were not processed prior to DNA extraction.

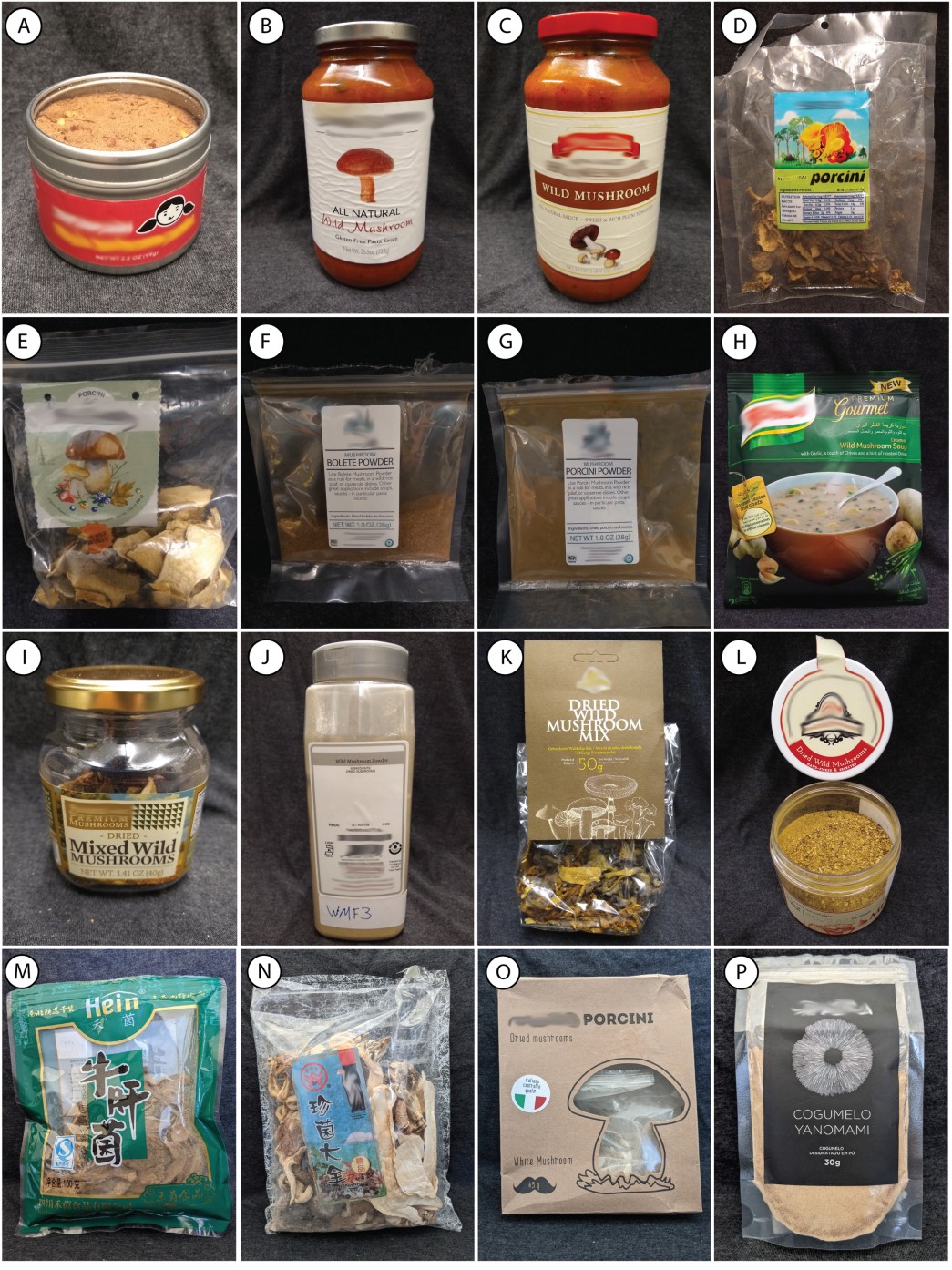

**Figure 1 Food products analyzed in this study.** (A) WMF-A. (B) WMF-B. (C) WMF-C. (D) WMF-D. (E) WMF-E. (F) WMF-F. (G) WMF-G. (H) WMF-H. (I) WMF-2. (J) WMF-3. (K) WMF-4. (L) WMF-5. (M) WMF-6 (N) WMF-10 (O) WMF-7 (P) WMF-11. Manufacturer names have been blurred to protect their identity using photo editing software.

DNA was extracted using the Zymo Research Quick-DNA Miniprep kit (#D3024, Zymo Research, Irvine, CA, USA). To enable efficient multiplexing of samples for metabarcoding, we employed a two-step amplicon protocol that first amplifies the marker

**Table 1 Sample, voucher, and species composition for each food product sampled for this study.** The format for Sample Designation is the project code (WMF) followed by the sample ID (A-H,1-11) followed by the DNA extraction replicate (1or 2) followed by the PCR replicate (1or 2) followed by the PCR replicate (a, b, or none). The number of ASVs assigned a species name by DADA2 is second from the right. Individual ASV numbers for ASVs that were not assigned a species name by DADA2 are at far right. Likely wild-collected species are indicated by an asterisk. Potentially poisonous species are in bold.

| Sample Designation | Accession Number | Stated Product Origin | Sample Type | Verbatim Ingredient Label | Scientific name for stated ingredients | Wild or Cultivable | Taxon assignment (DADA2) | Species ID (DADA2 & phylogeny) | Common Name | ASV counts | ASV Tree ID |
|---|---|---|---|---|---|---|---|---|---|---|---|
| WMF-A-1a | UT-M0001396 | Not stated | Mushroom Powder | Dehydrated porcini mushrooms | N/A | Wild | Agaricus sipapuensis | Agaricus bisporus | button mushroom | 44 | |
| | | | | | | | Suillus granulatus | **Suillus granulatus (SH1555178.08)\*** | slippery jack | 359 | |
| | | | | | | | Suillus brunnescens | Suillus luteus* | slippery jack | 41 | |
| | | | | | | | Unidentified Suillus sp. 1 | Suillus luteus* | slippery jack | | ASV 79, 134, 143, 145 |
| | | | | | | | Unidentified Suillus sp. 2 | Suillus aff. collinitus* | slippery jack | | ASV 47 |
| WMF-A-1b | UT-M0001396 | Not stated | Mushroom Powder | Dehydrated porcini mushrooms | N/A | Wild | Agaricus sipapuensis | Agaricus bisporus | button mushroom | 63 | |
| | | | | | | | Suillus granulatus | **Suillus granulatus (SH1555178.08)\*** | slippery jack | 960 | |
| | | | | | | | Lentinula edodes | Lentinula edodes | shiitake | 9 | |
| | | | | | | | Suillus pseudobrevipes | Suillus sp.* | slippery jack | 17 | |
| | | | | | | | Unidentified Suillus sp. 1 | Suillus luteus* | slippery jack | | ASV 112, 171, 182 |
| | | | | | | | Unidentified Suillus sp. 2 | Suillus aff. collinitus* | slippery jack | | ASV 125, 47 |
| WMF-A-2 | UT-M0001396 | Not stated | Mushroom Powder | Dehydrated porcini mushrooms | N/A | Wild | Suillus granulatus | **Suillus granulatus (SH1555178.08)\*** | slippery jack | 1806 | |
| | | | | | | | Agaricus sipapuensis | Agaricus bisporus | button mushroom | 93 | |
| | | | | | | | Lentinula edodes | Lentinula edodes | shiitake | 14 | |
| | | | | | | | Unidentified Suillus sp. 2 | Suillus luteus* | slippery jack | | ASV 176 |
| | | | | | | | Unidentified Suillus sp. 3 | Suillus aff. collinitus* | slippery jack | | ASV 22, 49 |
| WMF-B-1a | N/A | Western USA | Pasta sauce (chunks) | Mushrooms (portabella, shiitake, porcini and others) | Agaricus bisporus, Lentinula edodes, Boletus edulis | Wild and Cultivable | Suillus granulatus | **Suillus granulatus (SH1555178.08)\*** | slippery jack | 120 | |

| Sample Designation | Accession Number | Stated Product Origin | Sample Type | Verbatim Ingredient Label | Scientific name for stated ingredients | Wild or Cultivable | Taxon assignment (DADA2) | Species ID (DADA2 & phylogeny) | Common Name | ASV counts | ASV Tree ID |
|---|---|---|---|---|---|---|---|---|---|---|---|
| WMF-C-1a | N/A | Western USA | Pasta sauce (chunks) | Mushrooms (porcini, champignon, portabella) | Boletus edulis, Agaricus bisporus, Agaricus bisporus | Wild and Cultivable | Suillus granulatus | Suillus granulatus (SH1555178.08)* | slippery jack | 93 | |
| WMF-D-1a | UT-M0001397 | Not Stated | Dried Mushrooms | Porcini | N/A | Wild | Tylopilus microsporus | Tylopilus microsporus* | | 592 | ASV 4,22, 51, 59, 163, 8 |
| | | | | | | | Unidentified Retiboletus sp. 1 | Retiboletus fuscus (SH1505579.08)* | | | ASV 177 |
| | | | | | | | Unidentified Bothia sp. 1 | Tylopilus pseudoballouii (SH1612153.08)* | | | |
| | | | | | | | Unidentified Boletaceae sp. 1 | Unidentified Boletaceae sp. 1 (aff. SH1633670.08)* | | | ASV16, 148, 114, 21 |
| | | | | | | | Unidentified Boletaceae sp. 3 | Caloboletus yunnanensis (SH1516565.08)* | | | ASV 67, 98, 101, 88 |
| WMF-D-1b | UT-M0001397 | Not Stated | Dried Mushrooms | Porcini | N/A | Wild | Tylopilus microsporus | Tylopilus microsporus* | | 377 | ASV 129, 108, 127, 8, 4, 153 |
| | | | | | | | Unidentified Retiboletus sp. 1 | Retiboletus fuscus (SH1505579.08)* | | | |
| | | | | | | | Unidentified Boletaceae sp. 1 | Unidentified Boletaceae sp. 1 (aff. SH1633670.08)* | | | ASV 21, 16 |
| | | | | | | | Unidentified Boletaceae sp. 3 | Caloboletus yunnanensis (SH1516565.08)* | | | ASV 17, 20, |
| WMF-D-1c | UT-M0001397 | Not Stated | Dried Mushrooms | Porcini | N/A | Wild | Tylopilus microsporus | Tylopilus microsporus* | | 580 | ASV 59, 22, 136, 4, 90, 116, 103, 51, 8, 109, 115 |
| | | | | | | | **Amanita pseudoporphyria**\* | **Amanita pseudoporphyria**\* | | 6 | |
| | | | | | | | Unidentified Retiboletus sp. 1 | Retiboletus fuscus (SH1505579.08)* | | | ASV 152 |
| | | | | | | | Unidentified Bothia sp. 1 | Tylopilus pseudoballouii (SH1612153.08)* | | | |
| | | | | | | | Unidentified Boletaceae sp. 1 | Unidentified Boletaceae sp. 1 (aff. SH1633670.08)* | | | ASV 16, 21 |
| | | | | | | | Unidentified Boletaceae sp. 3 | Caloboletus yunnanensis (SH1516565.08)* | | | ASV 111, 113, 102, 149, 124, 95, 128, 17, 119, 20 |

| Sample Designation | Accession Number | Stated Product Origin | Sample Type | Verbatim Ingredient Label | Scientific name for stated ingredients | Wild or Cultivable | Taxon assignment (DADA2) | Species ID (DADA2 & phylogeny) | Common Name | ASV counts | ASV Tree ID |
|---|---|---|---|---|---|---|---|---|---|---|---|
| WMF-D-2 | UT-M0001397 | Not Stated | Dried Mushrooms | Porcini | N/A | Wild | Tylopilus microsporus | Tylopilus microsporus* | N/A | 468 | |
| | | | | | | | Lactifluus volemus | Lactifluus indovolemus (SH3565967.08)* | N/A | 198 | |
| | | | | | | | Amanita pseudoporphyria* | Amanita pseudoporphyria* | N/A | 25 | |
| | | | | | | | Unidentified Lactifluus sp. | Lactifluus indovolemus (SH3565967.08)* | N/A | | ASV 216 |
| | | | | | | | Unidentified Boletaceae sp. 3 | Caloboletus yunnanensis* | N/A | | ASV 68 |
| WMF-E-1a | UT-M0001398 | Siberia | Dried Mushrooms | Porcini | N/A | Wild | Boletus pinophilus | Boletus pinophilus* | porcini | 2325 | |
| WMF-E-1b | UT-M0001398 | Siberia | Dried Mushrooms | Porcini | N/A | Wild | Boletus pinophilus | Boletus pinophilus* | porcini | 2604 | |
| WMF-E-2 | UT-M0001398 | Siberia | Dried Mushrooms | Porcini | N/A | Wild | Boletus pinophilus* | Boletus pinophilus* | porcini | 3300 | |
| WMF-F-1 | UT-M0001399 | Not stated | Mushroom Powder | Dried bolete mushrooms | Boletus spp. | Wild | Agaricus sipapuensis | Agaricus bisporus | button mushroom | 121 | |
| | | | | | | | Suillus granulatus | Suillus granulatus (SH1555178.08)* | slippery jack | 467 | |
| | | | | | | | Lentinula edodes | Lentinula edodes | shiitake | 637 | |
| | | | | | | | Pleurotus ostreatus | Pleurotus ostreatus | oyster | 222 | |
| | | | | | | | Grifola frondosa | Grifola frondosa | maitake | 19 | |
| | | | | | | | Unidentified Suillus sp. 1 | Suillus luteus* | slippery jack | | ASV 25 |
| WMF-F-2 | UT-M0001399 | Not stated | Mushroom Powder | Dried bolete mushrooms | Boletus spp. | Wild | Pleurotus ostreatus | Pleurotus ostreatus | oyster | 361 | |
| | | | | | | | Suillus granulatus* | Suillus granulatus (SH1555178.08)* | slippery jack | 1258 | |
| | | | | | | | Agaricus sipapuensis | Agaricus bisporus | button mushroom | 127 | |
| | | | | | | | Lentinula edodes | Lentinula edodes | shiitake | 598 | |
| | | | | | | | Paxillus ammoniavirescens* | Paxillus ammoniavirescens* | N/A | 5 | |
| WMF-G-1 | UT-M0001400 | Not stated | Mushroom Powder | Dried porcini mushrooms | N/A | Wild | Boletus pinophilus | Boletus pinophilus* | porcini | 2832 | |
| WMF-H-1 | UT-M0001401 | Turkey | Powder Soup mix | Dried mushrooms, Mushroom powder, Mushroom granules | N/A | N/A | Agaricus sipapuensis | Agaricus bisporus | button mushroom | 4281 | |
| | | | | | | | Suillus granulatus | Suillus granulatus (SH1555178.08)* | slippery jack | 6 | |

| Sample Designation | Accession Number | Stated Product Origin | Sample Type | Verbatim Ingredient Label | Scientific name for stated ingredients | Wild or Cultivable | Taxon assignment (DADA2) | Species ID (DADA2 & phylogeny) | Common Name | ASV counts | ASV Tree ID |
|---|---|---|---|---|---|---|---|---|---|---|---|
| WMF-H-2 | UT-M0001401 | Turkey | Powder Soup mix | Dried mushrooms, Mushroom powder, Mushroom granules | N/A | N/A | Agaricus sipapuensis | Agaricus bisporus | button mushroom | 2843 | |
| | | | | | | | Agaricus bisporus | Agaricus bisporus | button mushroom | 636 | |
| | | | | | | | Unidentified Agaricus sp. 1 | Agaricus bisporus | button mushroom | | ASV 83, 116, 95, 149, 53, 85, 99, 6, 86, 144, 84, 141, 31, 112, 110 |
| WMF-1-1 | UT-M0001402 | not stated | Dried Mushrooms | boletus edulis, cantharellus, cibarius, leccinum | Boletus edulis, Cantharellus spp., Cantharellus cibarius, Leccinum spp. | Wild | Russula olivacea | Russula olivacea (SH1560124.08)* | | 78 | |
| WMF-2-1 | UT-M0001403 | Chile, China, Serbia, and/or Montenegro | Dried Mushrooms | Suilius luteus, Auricularia-judae, Boletus edulis, Pleurotus osteratus | Suillus luteus, Auricularia auricula-judae, Boletus edulis, Pleurotus ostreatus | Wild and Cultivatable | Pleurotus ostreatus | Pleurotus ostreatus | oyster | 2174 | |
| | | | | | | | Suillus brunnescens | Suillus granulatus (SH1555178.08)* | slippery jack | 158 | |
| | | | | | | | Unidentified Auricularia sp. | Auricularia aff. cornea | wood ear | | ASV67 |
| | | | | | | | Unidentified Suillus sp. 1 | Suillus luteus* | slippery jack | | ASV 37, 20, 10, 52, 56, 14, 9, 50, 23, 39 |
| WMF-2-2 | UT-M0001403 | Chile, China, Serbia, and/ or mentenegro | Dried Mushrooms | Suilius luteus, Auricularia-judae, Boletus edulis, Pleurotus osteratus | Suillus luteus, Auricularia auricula-judae, Boletus edulis, Pleurotus ostreatus | Wild and Cultivatable | Pleurotus ostreatus | Pleurotus ostreatus | oyster | 1378 | |
| | | | | | | | Suillus brunnescens | Suillus luteus* | slippery jack | 245 | |
| | | | | | | | Unidentified Suillus sp. 1 | Suillus luteus* | slippery jack | | ASV 139, 166, 140, 171, 24, 121, 107, 5, 63, 108, 46, 43, 155, 151, 162, 97 |
| WMF-3-1 | UT-M0001404 | not stated | Mushroom Powder | Dried mushrooms | N/A | N/A | Lentinula edodes | Lentinula edodes | shiitake | 912 | |
| | | | | | | | Suillus granulatus | Suillus granulatus (SH1555178.08)* | slippery jack | 641 | |
| | | | | | | | Agaricus sipapuensis | Agaricus bisporus | button mushroom | 168 | |
| | | | | | | | Paxillus ammoniavirescens* | Paxillus ammoniavirescens* | | 5 | |
| | | | | | | | Unidentified Retiboletus sp. 2 | Unidentified Boletaceae sp. 2 (aff. SH1152060.08)* | | | |
| WMF-4-1 | UT-M0001405 | Bulgaria | Dried Mushrooms | no ingredient list | N/A | N/A | Amanita caesareoides | Amanita caesarea* | Caesar's mushroom (a.k.a. ovolo (It.), ovolo impériale (Fr.)) | 254 | ASV71 |

(Continued)

| Sample Designation | Accession Number | Stated Product Origin | Sample Type | Verbatim Ingredient Label | Scientific name for stated ingredients | Wild or Cultivable | Taxon assignment (DADA2) | Species ID (DADA2 & phylogeny) | Common Name | ASV counts | ASV Tree ID |
|---|---|---|---|---|---|---|---|---|---|---|---|
| WMF-5-1 | UT-M0001406 | Europe | Mushroom Powder | no ingredient list | N/A | N/A | Calocybe gambosa | Calocybe gambosa* | St. George's mushroom | 90 | |
| | | | | | | | Boletus pinophilus | Boletus pinophilus* | porcini | 32 | |
| | | | | | | | Amanita caesareoides* | Amanita caesarea* | Caesar's mushroom (a.k.a. ovolo (It.), impériale (Fr.)) | 10 | |
| | | | | | | | Unidentified Auricularia sp. | Auricularia aff. cornea | wood ear | | ASV67 |
| | | | | | | | Unidentified Retiboletus sp. 2 | Unidentified Boletaceae sp. 2 (aff. SH1152060.08)* | N/A | | ASV71 |
| | | | | | | | Unidentified Suillus sp. 1 | Suillus luteus* | slippery jack | | ASV 50, 56, 37, 20, 23, 39, 52, 10, 9, 14 |
| WMF-5-2 | UT-M0001406 | Europe | Mushroom Powder | no ingredient list | N/A | N/A | Calocybe gambosa | Calocybe gambosa* | St. George's mushroom | 54 | |
| | | | | | | | Boletus edulis | Boletus edulis* | porcini | 5 | |
| | | | | | | | Unidentified Lactifluus sp. | Lactifluus indovolemus (SH3565967.08)* | N/A | | ASV 216 |
| | | | | | | | Unidentified Pleurotus sp. 1 | Pleurotus eryngii complex | King oyster | | ASV 209, 204 |
| | | | | | | | Unidentified Pleurotus sp. 2 | Pleurotus eryngii complex | King oyster | | ASV 190 |
| | | | | | | | Unidentified Agaricus sp. 1 | Agaricus bisporus | button mushroom | | ASV 6, 110, 99, 116, 144, 149, 83, 95, 85, 53, 141, 86, 84, 112, 31 |
| | | | | | | | Unidentified Boletaceae sp. 2 | Rugiboletus extremiorientalis* | N/A | | ASV 221 |
| | | | | | | | Unidentified Boletaceae sp. 3 | Caloboletus yunnanensis* | N/A | | ASV 68 |
| | | | | | | | Unidentified Suillus sp. 1 | Suillus luteus* | slippery jack | | ASV 162, 108, 43, 107, 76, 63, 139, 166, 5, 155, 151, 140, 46, 143, 97, 24, 121, 171 |
| | | | | | | | Unidentified Suillus sp. 2 | Suillus luteus* | slippery jack | | ASV 176, 114, 102, 103 |
| | | | | | | | Unidentified Suillus sp. 3 | Suillus aff. collinitus* | slippery jack | | ASV 49, 22 |

| Sample Designation | Accession Number | Stated Product Origin | Sample Type | Verbatim Ingredient Label | Scientific name for stated ingredients | Wild or Cultivable | Taxon assignment (DADA2) | Species ID (DADA2 & phylogeny) | Common Name | ASV counts | ASV Tree ID |
|---|---|---|---|---|---|---|---|---|---|---|---|
| WMF-6-2 | UT-M0001407 | Chengdu, Sichuan, China | Dried Mushrooms | Porcini Mushroom | N/A | Wild | Suillus granulatus | Suillus granulatus (SH1555178.08)* | slippery jack | 1872 | |
| | | | | | | | Grifola frondosa | Grifola frondosa | maitake | 5 | |
| | | | | | | | Suillus brunnescens | Suillus luteus* | slippery jack | 101 | |
| | | | | | | | Rutstroemia firma | Rutstroemia firma* | N/A | 60 | |
| | | | | | | | Auricularia auricula-judae | Auricularia auricula-judae | wood ear | 6 | |
| | | | | | | | Unidentified Suillus sp. 2 | Suillus luteus* | slippery jack | | ASV 102, 103, 114 |
| | | | | | | | Unidentified Suillus sp. 3 | Suillus aff. collinitus* | slippery jack | | ASV 22 |
| WMF-7-2 | UT-M0001408 | Russia | Dried Mushrooms | Dried White mushroom (Grade 1) | N/A | N/A | Boletus pinophilus | Boletus pinophilus* | porcini | 209 | |
| WMF-10-2 | UT-M0001411 | China | Dried Mushrooms | Mixed mushrooms (shiitake, oyster mushrooms, Nameko mushrooms) | Lentinula edodes, Pleurotus spp., Pholiota nameko | Wild and Cultivatable | Pleurotus ostreatus | Pleurotus ostreatus | oyster | 3130 | |
| | | | | | | | Lentinula edodes | Lentinula edodes | shiitake | 271 | |
| | | | | | | | Grifola frondosa | Grifola frondosa | maitake | 2011 | |
| | | | | | | | Pholiota microspora | Pholiota microspora | nameko | 307 | |
| | | | | | | | Unidentified Pleurotus sp. 1 | Pleurotus eryngii complex | king oyster | | ASV 209, 204 |
| | | | | | | | Unidentified Pleurotus sp. 2 | Pleurotus eryngii complex | king oyster | | ASV 190 |
| WMF-11-2 | UT-M0001412 | Brazil | Mushroom Powder | Favolus brasilliensis, Hydnopolyporus fimbriatus, Lentinula raphanica, Lentinus bertieri, L. concavus, L. crintius, Panus neostrigosus = P. lecomtei, P. strigellus, P. velutinus, Pleurotus albidus, P. djamor, Polyporus aquosus, P. phillippinesis, Polyporus aff thailandensis, P. tricholoma | Favolus brasilliensis, Hydnopolyporus fimbriatus, Lentinula raphanica, Lentinus bertieri, Lentinus concavus, Lentinus crintius, Panus neostrigosus, Panus strigellus, Panus velutinus, Pleurotus albidus, Pleurotus djamor, Polyporus aquosus, Polyporus phillippinesis, Polyporus aff. Thailandensis, Polyporus tricholoma | Wild | Lentinus striatulus | Lentinus striatulus* | N/A | 741 | |
| | | | | | | | Panus strigellus | Panus strigellus* | N/A | 956 | |
| | | | | | | | Lentinus berteroi | Lentinus berteroi/ crinitus* | N/A | 557 | |
| | | | | | | | Polyporus tricholoma | Polyporus tricholoma* | N/A | 268 | |
| | | | | | | | Ceriporia lacerata | Ceriporia aff. lacerata* | N/A | 157 | |
| | | | | | | | Pleurotus djamor | Pleurotus djamor* | pink oyster | 103 | |
| | | | | | | | Favolus tenuiculus | Favolus tenuiculus* | N/A | 335 | |
| | | | | | | | Podoscypha venustula | Podoscypha venustula* | N/A | 8 | |

gene using primers that include Nextera adapter tails, then using this as template for a second round of PCR to add unique indices and Illumina flow cell adapters (*Gohl et al., 2016*). First, the internal transcribed spacer 2 ("ITS2") region of the ribosomal RNA cistron was PCR-amplified using the primer pairs 5.8S-Fun_Nextera and ITS4-Fun_Nextera, which consist of the primers 5.8S-Fun and ITS4-Fun (*Taylor et al., 2016*) with Illumina Nextera adapter tails. Sequencing libraries were prepared from these amplicons by using a 1:99 dilution as template for a second round of PCR with indexing primers consisting of Nextera adapter, unique 8-bp index, and Illumina flow cell adapter. The resulting DNA was then cleaned and normalized across all samples using the AxyPrep Mag PCR Normalizer Protocol (Axygen Biosciences, Union City, CA, USA). Normalized DNAs were pooled and submitted for sequencing on an Illumina MiSeq with a Nano 250 Cycle Paired End Sequencing v2 flowcell at the University of Utah Genomics Core facility.

To investigate possible run and primer biases, a second sequencing run was performed using the same methods, but with independently prepared genomic extracts, different indexing primer combinations from the first run, and a negative control consisting of PCR reagents and water in place of gDNA. Additionally, three mushroom fragments from sample WMF-D were individually extracted and the full-length ITS regions were amplified using primers designed to preferentially amplify Agaricomycetes (*Dentinger, Margaritescu & Moncalvo, 2010*). Full-length ITS amplicons were depleted of excess primer and unincorporated dNTPs using ExoSAP-IT(CAT#78205.10.ML; ThermoFisher, Waltham, MA, USA) and submitted for Sanger sequencing on both strands to the University of Utah Sequencing Core. Sequences were edited and a consensus sequence created by overlapping both strands using Sequencher v5.4 (http://www.genecodes.com). Edited sequences were submitted to the National Center for Biotechnology Information (Accession#s MW979500 and MW979501).

## Data analysis

The DADA2 pipeline (*Callahan et al., 2016*) was used to generate a table of amplicon sequence variants (ASVs) from the raw Illumina data using the default parameters (Supplement 1). ASVs are similar to operational taxonomic units (OTUs) in that they represent biological variation, but are preferred to OTUs because they maximize the amount of biological information in the sequence data by not relying on arbitrary dissimilarity cutoffs to define them, making them reproducible outside of an individual dataset (*Callahan, McMurdie & Holmes, 2017*). DADA2 infers ASVs exactly and uses a training set to assign taxonomy to them. Each ASV was assigned a scientific name utilizing the näive Bayesian classifier (*Wang et al., 2007*) in DADA2 with the UNITE general release database v8.3 (10.05.2021) of all Fungi with singletons set as RefS as the training set (*Abarenkov et al., 2010*, *2021*). Further downstream analysis was performed by transferring sequences and metadata for each sample to the R package phyloseq (*McMurdie & Holmes, 2013*). For quality control and ease of analysis, species-level ASV assignments were collapsed based on taxonomy, with all assignments that had fewer than five counts removed. It is worth noting that DADA2 does not recognize the UNITE Species Hypothesis (SH) as a legitimate taxonomic rank. Therefore, in cases where the same
species name occurs in more than one SH, DADA2 is unable to distinguish between them. In many cases this may not be critical as the SHs will all be each other's closest relatives. However, in some cases the scientific names of species represented by more than one SH are polyphyletic. Therefore, we conducted phylogenetic analyses of ASVs assigned names by DADA2 as well as ASVs that could not be assigned at the species level to further clarify relationships to known taxa. Phylogenetic datasets were compiled by extracting all respective family- or genus-level sequences from the UNITE dataset v8.3 with global and 97% singletons, and adding to them representative ASVs from each sample for all taxa assigned by DADA2 at the species rank as well as all ASVs not assignable at the species level across all samples. For the Boletaceae, sequences from the genera *Chalciporus* and *Buchwaldoboletus* were included as the outgroup and for the Suillaceae, sequences of *Rhizopogon ochraceorubens* and *Rhizopogon hawkerae* were included as the outgroup. The datasets were reduced to the ITS2 region using ITSx (*Bengtsson-Palme et al., 2013*) and then multiple sequence alignments generated with the L-INS-i algorithm in MAFFT v7.475 (*Katoh et al., 2002*) using the online server (*Katoh, Rozewicki & Yamada, 2019*). Phylogenetic trees were inferred under maximum likelihood using IQ-TREE v2.0-rc1 (*Nguyen et al., 2015*) or the IQ-TREE web-server (*Trifinopoulos et al., 2016*) with the model of molecular evolution automatically determined by ModelFinder (*Kalyaanamoorthy et al., 2017*) and branch support estimated by ultrafast bootstrapping (*Hoang et al., 2018*). Trees are available in Supplement 2. Sequence data are available in the short-read archive of the INSDC (BioProject#-PRJNA642882). Alignments and trees are available in TreeBase (Study#-28291).

## RESULTS

After processing and quality filtering (including removal of sequences identified as species not known to produce macrofungi) DADA2 was able to assign species to a total of 51,766 ASV sequences, ranging from 59 to 571 per sample. Twenty-eight species were detected across all 16 food products, with compositions ranging from one to ten species of macrofungus per sample. In the seven samples for which we repeated DNA extractions (WMF-A, WMF-D, WMF-E, WMF-F, WMF-H, WMF-2, WMF-5) amplification and sequencing, the most abundant taxa detected in the first run were also the most abundant taxa detected in the second run. However, there were slight differences in low-abundance taxa detected in the two runs in five of the six samples. In WMF-5, four species detected in the first run were not detected in the second run, and seven species in the second run were not detected in the first run. However, the species with the highest number of ASVs (*Calocybe gambosa*) was the same in both runs. In sample WMF-A, one species was detected in the first run that was not detected in the second. In sample WMF-D, three species were detected in the first run that were not detected in the second, and one species was detected in the second run that was not detected in the first. In WMF-F, two taxa (*Suillus luteus*, *Grifola frondosa*) were detected only in the first run with moderate to low ASV counts (210 and 19, respectively), and one taxon (*Paxillus ammoniavirescens*) was detected only in the second run, but with a low ASV count (5). In WMF-H, *Suillus granulatus* was detected only in the first run with a low ASV count (5).

These run-to-run discrepancies likely indicate that although the species composition of the samples is diverse, they are often dominated by one or a few species and our sequencing depth was too modest to enable reproducibility of low-abundance taxa. The negative control only produced sequences in very low abundance that were removed completely during the quality filtering step used in DADA2. The most frequently encountered species in the samples were *Suillus granulatus* (in 10 samples), *Suillus luteus* (in eight samples) *Lentinula edodes* (in six samples), *Agaricus bisporus* (eight samples) followed by *Boletus pinophilus*, and *Pleurotus ostreatus* (in six and five samples, respectively).

Species were detected in nine samples that could not be assigned species-level identifications with DADA2. Most of these could be assigned to species based on phylogenetic results, except for an unidentified species of Boletaceae in sample WMF-D and a second unidentified species of Boletaceae in samples WMF-3 and WMF-5. Both of these unidentifiable species of Boletaceae were distinct from other sequences, but allied to sequences from unidentified species in the reference dataset. Species identified in most products did not agree with the contents listed on the labels (if usable identification was even present). Most species detected are considered edible, although the edibility of some species in WMF-D and WMF-F are suspect and that of the unidentifiable species cannot be evaluated. Independent Sanger sequencing of the gilled mushroom fragments identified in WMF-D confirmed the identifications from the metabarcoding runs, including *Amanita pseudoporphyria*. One fragment did not yield a readable DNA sequence.

## DISCUSSION

Sixteen different food products labeled as containing wild mushrooms were analyzed including dried mushrooms, mushroom powder, soup mixes, pasta sauces, and flavor enhancers. Many samples that were analyzed in this study contained edible mushrooms that are commonly collected and consumed, but only five of the products (WMF-E, WMF-G, WMF-2, WMF-7, WMF-11) had an ingredient label that accurately described the species composition of the contents. These included dried porcini mushrooms from Siberia (WMF-E, WMF-7) and one porcini powder with no origin statement (WMF-G), both of which contained *Boletus pinophilus*, one dried mushroom mix with multiple countries of origin (WMF-2), which contained the cultivated oyster mushroom (*Pleurotus ostreatus*) and wood ear (*Auricularia cornea*) as well as wild *Suillus luteus* (but not *Boletus edulis* as listed on the label), and the mushroom mix from the Yanomami in Brazil (WMF-11), which consisted of a variety of common tropical species not found in any other product (*Favolus tenuiculus*, *Podoscypha venustula*, *Panus strigellus*, *Lentinus* spp., *Polyporus tricholoma*, *Ceriporia* aff. *lacerata*, *Pleurotus djamor*). The latter species is a pantropical fungus that is widely cultivated and sold as the "pink oyster" in the U.S., but its presence in the mixture of wild mushrooms collected by Yanomami likely represents wild-harvested mushrooms. Species in the genus *Suillus* made up a substantial portion of the mushrooms found in the pasta sauces and the wild mushroom powders that were being sold as porcini (*Boletus edulis* and allies). Most of the products that claimed to include porcini consisted instead of widely cultivated species (button mushrooms, oyster

mushrooms, shiitake) and wild-collected *Suillus* spp. In fact, we were unable to identify any porcini species in all but two of the products labeled as containing this group of high-value wild, edible mushrooms. Interestingly, although *Suillus* spp. are widely traded and generally considered to be edible, the five species reviewed by (*Li et al., 2021*), which includes the two species we detected in our samples, are only listed as "E2: edible but with conditions". Some additional samples tested had no amplification, likely due to DNA degradation from harsh processing during food production (*e.g.*, heat, pressure, radiation) or inhibitory contaminants. This indicates the potential for there to be species present in samples that were not recovered by our methods. Unsurprisingly, the most recalcitrant samples were the pasta sauces, which are likely subjected to more extreme sanitizing conditions during production than other products. Yet, we were still able to recover sequenceable DNA from these as well as sample WMF-3, whose label specifically cites irradiation treatment, albeit with relatively low sequence counts.

Taxon assignment by DADA2 was less complete and sometimes inconsistent with taxon assignments that could be made from the phylogenetic results. For example, ASVs from WMF-5 were classified as *Amanita caesareoides*, but the phylogeny clearly indicates that these should be *Amanita caesarea* (Supplement 2). Similarly, some ASVs classified as *Agaricus sipapuensis* are phylogenetically allied to *Agaricus bisporus*. The widely cultivated *Agaricus bisporus* ("button mushroom") is far more likely than *A. sipapuensis*, the latter so far known only from two collections in montane coniferous forest in New Mexico. We also observed some differences in the assignment of *Suillus* spp. For example, in the phylogeny, there are two clades of ASVs assigned as either *Suillus* sp. NA or *Suillus brunnescens*, both of which appear to be very near or conspecific with *Suillus luteus*. Several species of Boletaceae found in samples WMF-D and WMF-5 that could not be assigned species-level names with DADA2 could also be assigned species names based on the phylogenetic results. These discrepancies suggest that DADA2 did not perform well with this dataset, perhaps due to the relatively poor resolving power of the ITS2 region by itself, and highlights the importance of careful scrutiny of results from automated pipelines like DADA2. As in other studies that have shown the failure of automated pipelines to accurately represent fungal diversity in complex samples (*e.g.*, (*Hoang et al., 2018*; *Hofstetter et al., 2019*)), in this study phylogenetic-based taxon assignment was necessary to accurately describe the contents of our samples.

Some dried mushroom products contained a mixed collection of species that were not accounted for on the list of ingredients. For example, the ingredient list for sample WMF-6 was "porcini mushroom", yet the species we detected included the wild-collected species *Suillus granulatus* and *S. luteus*, the likely cultivated species *Grifola frondosa* (maitake) and *Auricularia auricula-judae* (wood ear), and *Rutstroemia firma*, a lesser known wood-dwelling ascomycete fungus that produces fleshy sporocarps of unknown edibility. This latter fungus is an unusual component of a wild mushroom food mixture and, if not introduced intentionally, may be due to accidental misidentification of wild-collected and superficially similar-looking taxa (*e.g.*, *Auricularia*), or possibly as a contaminant of the growth substrate used to cultivate maitake or wood ears.

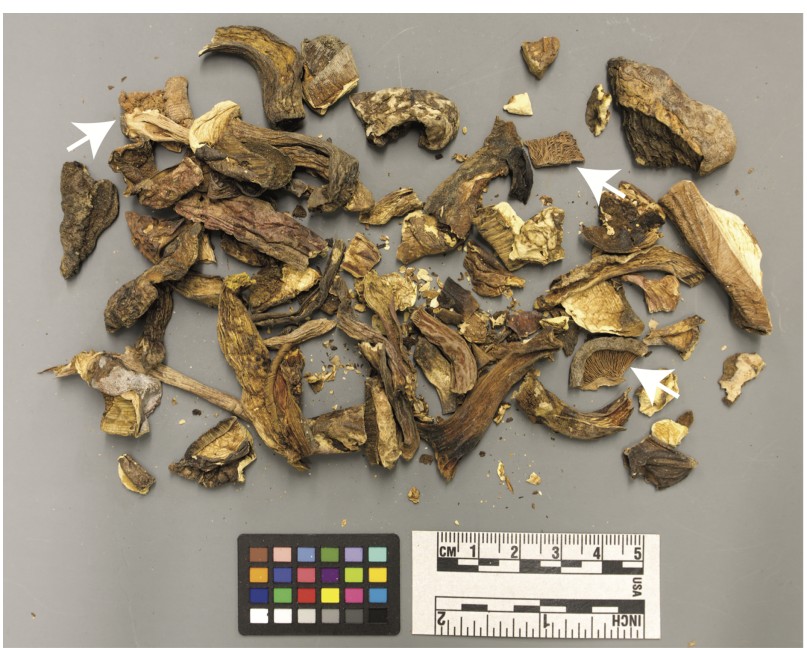

**Figure 2  Contents of WMF-D.** Arrows indicate gilled mushroom fragments. Image courtesy of Natural History Museum of Utah (UMNH), UT-M0001397.

Sample WMF-D had the second most diverse composition of macrofungi, none of which were indicated by the package labelling. Although WMF-D was labeled as porcini (non-gilled mushrooms with a spongy layer under the cap), a cursory visual inspection revealed fragments of gilled (lamellate) mushrooms, visible even through the packaging (Fig. 2). Among the species we could identify were the non-lamellate boletes *Tylopilus microsporus*, *Tylopilus pseudoballouii*, *Retiboletus fuscus*, and *Caloboletus yunnanensis*, and the lamellate agarics *Lactifluus indovolemus* and *Amanita pseudoporphyria*. All of these species are known to occur only in Asia and only very limited data are available on their edibility outside of local knowledge. We also detected a fifth species of Boletaceae that we could not assign to sequences in the reference dataset, indicating that they may represent poorly known species that may even be new to science, as reported earlier for porcini exported from China (*Dentinger & Suz, 2014*). Perhaps most alarmingly, one of the gilled mushrooms sampled from WMF-D was identified as *Amanita pseudoporphyria* (a.k.a. "Hongo's False Death Cap"), a species belonging to the group containing the most notoriously poisonous mushrooms such as the "Death Cap" (*A. phalloides*) and the "Destroying Angels" (including *A. virosa*, *A. bisporigera*, *A. ocreata*, and others). Other species of *Amanita* are well-known to be edible and even highly-valued, such as *Amanita caesarea* (found in sample WMF-5), yet only one of the 12 species recently reviewed for edibility was confirmed as edible (*Li et al., 2021*) . Although listed as "E2: edible but with conditions" by *Li et al. (2021)*, and apparently not an uncommon species in wild mushroom markets in Yunnan province in China (*Wang, Liu & Yu, 2004*), consumption of *Amanita pseudoporphyria* has been shown to cause renal failure in humans (*Iwafuchi et al., 2003*). A review of mushroom poisonings over an 18 year period in southern

China reported that ingestion of *Amanita* spp. was responsible for over 78% of people poisoned by mushrooms and over 70% of fatalities due to mushroom poisoning (*Chen, Zhang & Zhang, 2014*). *Chen, Zhang & Zhang (2014)* specifically reported records of poisonings by mushrooms identified as *Amanita* cf. *pseudoporphyria* in 22 people, four of which died. Perhaps unsurprisingly, several customers left reviews of this product describing the mushrooms as having "an extremely bitter flavor and a bad aftertaste", or claiming that the mushrooms "made those that ate it violently ill" or that they were "poisoned" and had "never been so sick in my life". One even reported that "the package I rec'd included a mashed cigarette butt". Upon realizing the packet contained potentially poisonous mushrooms, in July 2019 we contacted the online retailer from where this product was purchased and informed them of our findings. As of the time of this writing, this product remains for sale.

The only other sample in which we detected potentially poisonous species was WMF-F, which had *Paxillus ammoniavirescens*, a species closely related to the 'Poison Paxillus' (*Paxillus involutus*). *Paxillus involutus* has been implicated in acquired hypersensitivity leading to renal failure and immunohemolysis following consumption (*Flammer, 1985*). Although no direct edibility data were available for *Paxillus ammoniavirescens*, *Li et al. (2021)* list *P. involutus* as E2 despite the documented poisonings. However, this species was only detected with a very low ASV count (5) and in only one of two replicated sequencing runs, so its presence could be a technical artifact. Nonetheless, it suggests that there may be low levels of this species intermixed with more traditional edible taxa and even small amounts of it could be problematic for people that may be especially sensitive to them or have acquired this hypersensitivity after consuming them a variable and undetermined number of times. The species composition of these samples found to contain potentially poisonous mushrooms is somewhat alarming. Presence of known toxin-producing species suggests the need for greater scrutiny of wild mushroom food products to avoid inadvertent poisonings. Why would potentially toxic species be encountered in globally distributed wild mushroom food product? Although negligence or fraud cannot be dismissed, the other likely explanation comes from the simple truth that fungal diversity is still very poorly documented. Given our poor state of knowledge of fungi, it may not be such a surprise if it is discovered that in under-documented regions there are poisonous species that are frequently confused with edible species, or that the presence of toxins can vary from sample to sample in ways that are currently not known.

We also detected microfungi (molds and yeasts) in addition to macrofungi, although most of these were not well-represented in the sequencing runs. These included species of the molds *Penicillium*, *Hypomyces* and *Cladosporium*, and the yeasts *Candida*, *Dipodascus*, *Vanrija*, *Aureobasidium*, *Cryptococcus* and *Wallemia*. While some of these are expected environmental contaminants, others are likely directly associated with the fresh mushrooms as parasites (*Hypomyces boletiphagus*, *H. chlorinigenus*, *H. perniciousus*, and *Dipodascus armillariae*) or as commensals (*Vanrija humicola*). The contribution of these contaminating organisms to potential adverse effects following ingestion are unknown but may be worth considering in future studies, especially because some of these are known to produce potent bioactive chemicals (*e.g.*, *Penicillium*).

## CONCLUSIONS

This study is a preliminary survey of the species composition in food products labeled as containing "wild mushrooms" sold in the USA. Our results indicate that label inaccuracies are widespread and that many products claiming to contain wild mushrooms instead contained only commonly cultivated species of mushrooms such as white button/ portabella (*Agaricus bisporus*), oyster (*Pleurotus* spp.), and shiitake (*Lentinula edodes*), as well as wild-collected species with inferior culinary quality such as slippery jacks (*Suillus* spp.). Although our results demonstrate a severe lack of transparency, most of the mushrooms identified in the food samples are popular varieties of edible taxa with no indication of potential adverse health effects. However, the presence of some fungi that are potentially poisonous to humans demonstrates the potential for toxic species to enter the international food supply chain with little or no oversight. Moreover, the presence of gilled mushrooms mixed in packets labeled as porcini demonstrates a clear lack of quality control in the preparation of one product (WMF-D). In addition to the potential negative health impacts of fraudulent food labeling, the poor alignment of the contents with what is listed in the ingredients means it is impossible to regulate trade of wild collected mushrooms in the international food supply chain, potentially resulting in unsustainable harvesting practices that may put rare and threatened species at risk of extinction. A more thorough review of the contents of wild mushroom food products should be conducted to establish how extensive the problem is.

## ACKNOWLEDGEMENTS

We are grateful to Bryce Alex for help with data anaylsis, to Giuliana Furci for supplying us with the Yanomami mushrooms and for a review of an earlier draft of the manuscript, and to Keaton Tremble and two anonymous reviewers for comments on the manuscript.

### Funding

The authors received no funding for this work.

### Competing Interests

Bryn Dentinger is an Academic Editor for PeerJ.

### Author Contributions

- W. Dalley Cutler II conceived and designed the experiments, performed the experiments, analyzed the data, prepared figures and/or tables, authored or reviewed drafts of the paper, and approved the final draft.
- Alexander J. Bradshaw performed the experiments, analyzed the data, prepared figures and/or tables, authored or reviewed drafts of the paper, and approved the final draft.
- Bryn T.M. Dentinger conceived and designed the experiments, analyzed the data, prepared figures and/or tables, authored or reviewed drafts of the paper, and approved the final draft.

## DNA Deposition

The following information was supplied regarding the deposition of DNA sequences:

The ITS2 amplicon sequences are available at NCBI's Short Read Archive: PRJNA642882.

The alignments and trees are available at Figshare: Dentinger, Bryn (2021): Wild mushroom food authentication. figshare. Dataset. DOI 10.6084/m9.figshare.14959842.v1.

## Data Availability

The ITS2 amplicon sequences are available at NCBI's Short Read Archive: PRJNA642882.

## Supplemental Information

Supplemental information for this article can be found online at http://dx.doi.org/10.7717/peerj.11747#supplemental-information.

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
