# Peer review of "What’s for dinner this time?: DNA authentication of “wild mushrooms” in food products sold in the USA"

_PeerJ, doi:10.7717/peerj.11747_

## Round 0.1 · original submission · Major Revisions

Kindly address the comments made by reviewers. They find your manuscript very useful to advance knowledge in the area. Look forward to your revised manuscript.

Reviewer 1 ·

Basic reporting

- very well written

- the references are good overall although I provided a few additional references for the authors to consider. some DNA barcoding has been done for medicinal fungi and medicinal plants, which may be worth mentioning (since the sample problems were found in those studies)

- very professional and well done

- since it was descriptive there were not truly hypotheses but in this case that seemed appropriate

Experimental design

- research question was well defined and certainly this is important for food safety and for conservation

- rigorous investigation although number of reads for the next gen sequencing was low. I made some comments on the manuscript regarding the fact that it would be good to see information on negative controls and any potential contamination sources (see PDF)

Validity of the findings

- certainly this is relevant information and it has not been previously assessed in this way. it is definitely complimentary to other findings in medicinal plants, fungi, and other examples given in the intro on sushi and meat products
- underlying data seem solid and the only exception is more info needed on sequence processing and potential contamination issues (see above)
- overall very interesting work and the finding of a potentially toxic species in edible fungi packets being sold commercially is concerning and should grab the attention of regulators

Additional comments

I enjoyed this paper and think it is a solid and useful contribution. Please see detailed comments on the PDF that may help in the final revisions.

Annotated reviews are not available for download in order to protect the identity of reviewers who chose to remain anonymous.

Reviewer 2 ·

Basic reporting

In general well written, well explained pilot study to show the problems with the lack of labeling on mushroom products.
The strength lies in the fact that this study has been undertaken, and that even in this small sampling (only 12 products were tested), the results are surprising and scary.
The weakness is in the small sampling size, and the absence of standard sampling of the products; how many mushroom pieces were used for each product and how were they chosen? A second weakness is the use of only ITS2 for identification; the whole ITS region is considered the fungal barcode gene region, but in some groups ITS2 lacks the same level of specificity.

This is actually not the first time a study of commercial food products labled as contaning fungi is conducted; see Albertsen in Svampe 2 (1980) on truffles in food (in Danish). [http://www.svampe.dk/svampe/SVAMPEpdf/svampe2.pdf]

The introduction lays the ground work for the issues that are investigated, the emphasis in the article should be on the USA; the EU and separate EU countries have their own legislation which is in many cases more precise than that in the USA.
It would be helpful to clearly indicate which information at the moment is mandatory in the USA - for instance, is country of origin required ? is a list of species required for imported products? and if so should they be common names or scientific names ? Are there differences between the sales of fresh wild-collected mushrooms and mushrooms that are dried or are processed (this in particular is not very clear in the introduction).
How do the regulations for herbal products differ or not from those for mushroom products?
The stress in this article is on the poisonous mushrooms, but many people experience unpleasantness after eating species that are not considered poisonous, but for which they are sensitive. More emphasis on this would be welcome.
It feels artificial to include some extra sentences on habitat degradation and decline of harvested species (in introduction and conclusions). This is beyond the scope of this article, here it is about “is what is on the lable indeed in the product?”

For readers who are not familiar with which species are cultivated and which ones are wild, it is necessary to indicate that in Table I, and also to translate the common names used on lables into scientific names, as used in the results column, in some cases that should be a genus name, such as for ‘porcini’ and ‘woodear’.

Also in Table I the contents of WMF2 are given with Latin names -common names are listed on the product, and these are in fact quite accurate (only porcini were not found in the product)

the English is good to follow, but there are a few awkward sentences and grammatical mistakes (awkward wording at lines 14–15 and 42-43; line 48, 80 - should be plural instead of singular) Table I WMFD contains Pulchroboletus rubricitrinus.

References for the effects of Paxilus involutus and allies should be for instance the following (not a guide book!):
Flammer, R. Paxillus syndrome: immunohemolysis following repeated mushroom ingestion. Schweizerische Rundschau für Medizin Praxis 74, 997–999 (1985).
Bschor, F., Kohlmeyer, J. & Mallach, H. J. Neue Vergiftungsfälle durch Paxillus involutus. Zeitschrift für Pilzkunde 29, 1–3 (1963).
Winkelmann, M., Stangel, W., Schedel, I. & Grabensee, B. Severe hemolysis caused by antibodies against the mushroom Paxillus involutus and its therapy by plasma-exchange. Klinische Wochenschrift 64, 935-938, doi:10.1007/bf01728620 (1986).

references -
- could you check those for Gelardi and (use capital letters for genus names),and Zhao (capital letters, pages 1127–1136)
- for Von Hagen et al. i found the following: Von Hagen, B., Weigand, J. F., McLain, R., and Fight, R. (1996). Conservation and development of nontimber forest products in the Pacific Northwest: an annotated bibliography. Gen. Tech. Rep. PNW-GTR-375. Portland, OR: U.S. Department of Agriculture, Forest Service, Pacific Northwest Research Station. 246 p. DOI: 10.2737/PNW-GTR-375
- Lowenstein et al. : PLoS ONE 4(11): e7866.
- is there a web site for “Nair, M. P. (2016). National survey of state regulation of wild mushroom foraging for retail sale. AFDO Board of Directors , page 5.”? I failed to find this.

Experimental design

see above under strengths and weaknesses

Validity of the findings

For the results - the name Suillus granulatus is used for several different species, could you specify which ones were found in the products that were sampled.
In line 155 it is stated that WMFE contains a Paxillus species - i assume WMFA is meant.
and in Figure 2 - the fragment in the upper left hand part of the picture also looks like a gilled mushroom (a Russula or an Amanita species).
Could you, based on the results of your identifications, say from which geographical area the products came from? I only saw one country of origin (Bulgaria for WMF4).

---

## Round 0.2 · Minor Revisions

Author, please kindly attend to the reviewer's comments. Looking forward to your revised manuscript.
Thank you very much.

Reviewer 2 ·

Basic reporting

In general my comments are minor and have to do with clearness of the text, nuancing and extra information. These are indicated in the pdf file.

I recommend structuring the methods section, and starting out saying that three different methods were used to extract and sequence the mushroom bits, followed by an elaboration of these methods.

The sample designation in table 1 is quite unclear, and could be explained more clearly in the legend. Are the replicates done with the same methods or different ones? and what doe the a and b annotations stand for?

The list of references needs your attention, as capital letters have been changed to lower case (e.g. china instead of China, and its for ITS).

Experimental design

the only comment i have is on the sampling and mixing of the samples (see the pdf); i recommend more clarity in this section.

Validity of the findings

I recommend that the authors read more about mushroom poisonings, and assumed mushroom poisonings in European mycological journals; for instance in Germany there were cases of severe poisonings by people eating Chinese dried boletes which contained parts of leaves of Araceae (perhaps used for packaging); these were the culprits for the poisonings, and not the mushrooms.

As for protection of mushrooms in the USA, not a single mushrooms species is included in the Endangered species act, but regionally, they are included in conservation efforts, e.g. in the Northwest Pacific Forest Plan. And globally a number of fungal species is now listed in the IUCN red data list, but that list is an advocacy tool. These include species that are local in the USA.

Additional comments

Thank you for incorporating the comments and requests made by the reviewers.

In general my comments are minor and have to do with clearness of the text, nuancing and extra information. These are indicated in the pdf file.

my other recommendations are in the two boxes above and in the text itself.

Annotated reviews are not available for download in order to protect the identity of reviewers who chose to remain anonymous.

---

## Round 0.3 · accepted · Accept

Thank you authors for revising your work, and addressing all concerns raised. The authors have benefited from the peer-review process. The revised manuscript is now acceptable for publication. Thank you for finding PeerJ as your journal of choice, and looking forward to your future scholarly contributions.

Congratulations, and very best wishes.